# In Vitro Nano-Polystyrene Toxicity: Metabolic Dysfunctions and Cytoprotective Responses of Human Spermatozoa

**DOI:** 10.3390/biology12040624

**Published:** 2023-04-20

**Authors:** Martina Contino, Greta Ferruggia, Stefania Indelicato, Roberta Pecoraro, Elena Maria Scalisi, Giovanni Bracchitta, Jessica Dragotto, Antonio Salvaggio, Maria Violetta Brundo

**Affiliations:** 1Department of Biological, Geological and Environmental Sciences, University of Catania, 95124 Catania, Italy; greta.ferruggia@phd.unict.it (G.F.); stefaniaindelicato93@gmail.com (S.I.); roberta.pecoraro@unict.it (R.P.); elenamaria.scalisi@unict.it (E.M.S.);; 2U.O. Fisiopatologia della Riproduzione Umana—Clinica del Mediterraneo, 97100 Ragusa, Italy; bracchitta@centroaster.com (G.B.); jessica.dragotto@gmail.com (J.D.); 3Experimental Zooprophylactic Institute of Sicily “A. Mirri”, 90129 Palermo, Italy; antonio.salvaggio@izssicilia.it

**Keywords:** human spermatozoa, polystyrene, nanoplastics, oxidative stress, DNA fragmentation, protective responses

## Abstract

**Simple Summary:**

The question of emerging pollutants, among which nanoplastics occupy a predominant position, requires further investigation regarding their interaction with different biological systems, including reproductive cells. In the present evaluation, deleterious effects on sperm cells have been correlated with polystyrene nanoparticle size. Indeed, the decline of fundamental parameters such as motility, acrosome and DNA integrity, and ROS physiological production, has been associated with the action of nanoparticles with a diameter of 50 nm, compared to particles with larger size (100 nm). In addition, the expression of protective biomarkers, such as HSP70s, has been shown to contribute to damage recovery. The results warn about persistent and chronic contamination of plastics with a focus on infertility to elucidate the metabolic and biochemical changes of sperm cells in the presence of stressful xenobiotics.

**Abstract:**

The ubiquitous spread of Polystyrene nanoplastics (PS-NPs) has rendered chronic human exposure an unavoidable phenomenon. The biodistribution of such particles leads to bioaccumulation in target organs including the testis, the site of sperm maturation. The purpose of this research has been to estimate the impact of PS-NPs (50 and 100 nm) on the metabolism of mature spermatozoa. The analysis of the semen parameters has revealed a higher toxicity of the smaller sized PS-NPs, which have negatively affected major organelles, leading to increased acrosomal damage, oxidative stress with the production of ROS, DNA fragmentation, and decreased mitochondrial activity. PS-NPs of 100 nm, on the other hand, have mainly affected the acrosome and induced a general state of stress. An attempt has also been made to highlight possible protective mechanisms such as the expression of HSP70s and their correlation among various parameters. The results have evinced a marked production of HSP70s in the samples exposed to the smaller PS-NPs, negatively correlated with the worsening in oxidative stress, DNA fragmentation, and mitochondrial anomalies. In conclusion, our results have confirmed the toxicity of PS-NPs on human spermatozoa but have also demonstrated the presence of mechanisms capable of counteracting at least in part these injuries.

## 1. Introduction

The spermatozoon is the highly specialized and differentiated cell whose peculiarities are decisive to complete the fertilization process. Its metabolism is finely regulated, through sophisticated adaptations, to maintain intact all the essential components of the cell for oocyte encounter and zygote formation [1,2]. For instance, its peculiar shape ensures proper progressive movement, the presence of the acrosomal vesicle guarantees the opening of a passage in the egg cell envelopes, the highly condensed DNA protects the genetic heritage, and the plastic membrane can withstand the fluidity changes required during the passage and transport in the epididymis and subsequent capacitation in the female tract. These are only some of the physiological properties of a healthy spermatozoon, but many others can be mentioned, such as the maintenance of proper energy intake by the mitochondrial helical sheath or proper activation of the signaling pathways [3]. Thus, the appropriate cooperation of the various organelles and the correct consequentiality of events are required to obtain conception. It is well known that any dysfunction at the structural, biochemical, molecular, or metabolic level, because of disease, infection, or contact with contaminants, may affect reproductive success and cause infertility. Furthermore, the proven sensitivity to pollutants makes the sperm cell a good indicator of reproductive risks related to xenobiotic pollution, which is currently dominated by the massive spread of PS-NPs [4]. 

Since 1930, global annual production of Polystyrene (PS) has exponentially grown due to its low affordability, ease of fabrication, versatility, thermal efficiency, and moisture resistance, so it finds multiple applications in a variety of fields, especially for the synthesis of industrial products (creams, toys, food packaging) [5]. It is estimated that 21 million tons of PS, which corresponds to 7.1% of all plastic present on the globe, are manufactured annually [6]. Over the decades, scientists have experimented with supplementing substances to ensure the resistance of PS products. Longevity is the primary cause of its persistence in several environmental matrices. In fact, PS is a durable thermoplastic that has a very low degradation rate in the natural environment, so it remains as a solid waste in nature for a long time. Despite this, the slow degradation has led to the diffusion of micro- and nanoscale fragments capable of penetrating into organisms. As has been widely demonstrated, PS-NPs are absorbed and distributed in the tissues of various animals, including mammals (cattle, sheep, etc.) [7,8,9]. Recently, these mechanisms have also been observed in humans. In a study conducted by Leslie et al. (2022), PS-NPs have been found at the level of the bloodstream. Thus, even in humans, as already demonstrated in other mammals, NPs could move from the bloodstream into the target organs. [10]. For example, traces of NPs have been reported in mice at the level of the liver, kidney, and intestine [11,12]. It is also known that the reproductive system is one of the most vulnerable to the presence of pollutants [13]. Jin et al. (2021) have, in fact, demonstrated how the testis is a target for nanoplastics that can reach the lumen of the seminiferous tubules through disruptive phenomena of the hematotesticular barrier, directly impairing the microstructure of spermatozoa [14].

In this sense, the aim of the present study has been to evaluate, following 30 min of exposure, the effects of polystyrene nanospheres with different diameters (50 and 100 nm) on the metabolism of human spermatozoa, analyzing several semen parameters and possible cytoprotective strategies in response to the injury.

## 2. Materials and Methods

### 2.1. Materials

Polystyrene nanospheres were purchased from Sigma-Aldrich (St. Louis, MO, USA), along with the other solutions necessary for the conduct of this experiment (PBS, 2′,7′-dichlorofluorescein-diacetate, DMSO, BSA, DAB). Human sperm culture and washing medium were obtained from FertilPro (Industriepark Noord, Beernem, Belgium). Eosin Y and Methylene blue and Formaldehyde were purchased from Bio-Optica (Milan, Italy). Peanut Agglutin-Fluorescein solution was obtained from Vector Laboratories (Newark, CA, USA). Fluoromount G with DAPI and Hoechst 33342 were purchased from Invitrogen (Waltham, MA, USA). A Halosperm Kit was obtained from Halotech (Madrid, Spain). HSP70 primary antibody and goat Anti-rabbit IgG pre-adsorbed Rhodamine secondary antibody were obtained from GeneTex (Irvine, CA, USA). Triton X-100 was purchased from ChemSolute (Renningen, Germany). Finally, a MiOXYS system was obtained from Medical Biological Technologies (MBT, Pretoria, South Africa). 

### 2.2. Preparation of Solutions

Amino-modified polystyrene nanospheres with diameters of 50 and 100 nm were purchased from Sigma-Aldrich. These emit, upon excitation, a blue (excitation/emission: 358/410 nm) and orange/green (excitation/emission: 481/644 nm) fluorescence, respectively. 

From the stock solution, consisting of PS-NPs in aqueous solution at a concentration of 1.06–1.07 g/mL, the following concentrations were prepared in human sperm culture medium (FertilPro): 0.1 µg/mL, 0.5 µg/mL, 1 µg/mL. To avoid the formation of aggregates, the solutions were sonicated for 2 min in a sonicator (Sonoplus). Given the lack of studies on human spermatozoa in the literature, these concentrations have been selected in accordance with previous studies conducted in mice [15,16,17]. Based on them, NPs at these concentrations cross the blood–testicular barrier and interact directly with spermatozoa. The selected concentrations could be higher than the real concentrations to which human spermatozoa might be exposed, but the aim of the experiment was to identify effective concentrations.

### 2.3. Collection of Human Spermatozoa

Semen samples were received at the laboratory of Biotechnology of Reproduction—Department of Biological, Geological and Environmental Sciences, University of Catania, Italy, from the PMA (Medically Assisted Procreation) Center of Medi. San s.r.l.—Clinica del Mediterraneo, Ragusa, Italy. 

Semen samples (n = 8) were collected, in the year 2023, and delivered by the patients according to the standard protocols set by the WHO Manual for the Examination and Analysis of Seminal Fluid (WHO 2021) [18]. Subsequently, careful analysis by spermiogram assessed, both macroscopically (volume, pH, appearance, fluidization, and viscosity of seminal fluid) and microscopically (number, viability, motility, morphology of human spermatozoa), the suitability of these samples for experimental purposes. Normospermic samples with more than 70% motile spermatozoa were selected. Finally, semen samples were cryo-frozen in liquid nitrogen at −196 °C and thawed as needed. The semen characteristics of normospermic individuals whose samples were involved in the present study are shown in Table 1. 

### 2.4. Informed Consent

Patients of the PMA Center who donated their sperm for research purposes signed the informed consent and anonymity was guaranteed. The study was conducted according to the criteria of the Declaration of Helsinki (2001) [19]. The PMA Center-Mediterranean Clinic (Ragusa, Italy) is authorized for cell donation, supply, testing, processing, preservation, storage, and distribution by the National Transplant Center (NTC) and the Superior Institute of Health under ministerial code 190002. The patient authorizations are included in the Appendix A.

### 2.5. Thawing and Exposure

The thawing of samples was executed following the standardized protocol of the WHO Manual (WHO 2021) [14]. Samples were placed for 10 min at room temperature and for 10 min at 37 °C with 5% CO_2_ in a water bath. To verify that thawing had taken place correctly and that the spermatozoa had regained the ability to move, 10 µL of sample was placed in a Neubauer counting chamber and observed under a light microscope at 40× magnification. Furthermore, because cryopreservation and thawing result in the loss of sperm viability from 50.6% to 30.3% [18], samples with ≥40% of viability were used in the experiment. To separate motile spermatozoa from immobile spermatozoa, the direct swim-up procedure, without centrifugation, was carried out by placing the tubes at a 45° inclination for 1 h in a 37 °C incubator with 5% CO_2_. Next, the samples were aliquoted, drawing the surface part of the sample corresponding to the fraction of spermatozoa with high motility, into four Eppendorf tubes: three for exposure for both nanoplastics and one for control (CTRL or unexposed). At this point, to separate the semen from the plasma, a cleavage was conducted, adding washing medium (FertilPro) in a 1:2 ratio, and centrifuging for 10 min at 2000 rpm. 

After discarding the supernatant, the pellet was resuspended on Eppendorf tubes: in the case of the six exposed samples, with 450 µL of culture medium containing the PS-NPs at different concentrations (0.1–0.5–1 µg/mL) and with different diameter; in the case of the control, with 450 µL of only culture medium. The toxicity test involved the exposure of the gametes for 30 min at 37 °C with 5% CO_2_ to assess the different basic parameters of the semen. For each test, at least 200 (~400) spermatozoa were counted in 5 different random fields, and 2 replicates were performed.

### 2.6. Evaluation of Motility

Measurement of the different parameters describing sperm movement was carried out by video analysis in the ImageJ program, in which the CASA plugin, devised by Wilson-Leedy and Ingermann (2007) [20] for the analysis of Zebrafish sperm motility, was installed. The CASA system evaluated the parameters, as shown in Table 2.

In the present experiment, following exposure of the samples to the PS-NPs, 10 µL of sample was placed on a slide and covered with a 24 × 24 coverslip. The slides were examined under a light microscope at 10× magnification after approximately 30 s to allow for stabilization of the fluid within the chamber created by the placement of the coverslip. Videos of 10 s each were recorded via a video camera (Nikon Y-TV55, Amsterdam, Netherlands) connected to the microscope. After uploading the videos into ImageJ in .avi format, the images were adjusted to mark the sperm heads in red on a white background. The CASA plugin has also been adapted for motion analysis of other animal species, including mammals, such as stallions [21]. At this point, the CASA plugin was run, entering the values shown in Table 3.

### 2.7. Evaluation of Plasma Membrane Integrity

Plasma membrane integrity was assessed by a colorimetric assay using Eosin Y (0.5%, *w*/*v*, Bio-Optica) [22]. An amount of 10 µL of the sample was placed on the slide, to which 10 µL of dye was added. The slide was covered by the coverslip and read under a light microscope (Nikon Eclipse E-200, Amsterdam, Netherlands) at 40× magnification. Spermatozoa with compromised membranes were stained pink as the dye penetrated inside the damaged and dead cells, while viable spermatozoa were transparent.

### 2.8. Assessment of Acrosomal Vesicle Integrity

Acrosome status was analyzed following the protocol described by Lybaert et al. (2009), with some modifications [23]. An amount of 10 µL of the sample was smeared onto a slide and allowed to air dry. The sample was then fixed in methanol for 15 min at 37 °C. Following 3 washes in Phosphate Buffered Saline (PBS, Sigma-Aldrich), the slides were incubated with a Peanut Agglutin-Fluorescein solution (PNA-FITC, Vector Laboratories Newark, CA, USA) with a concentration of 10 µg/mL for 30 min at 37 °C in the dark. At the end of incubation, slides were washed in PBS and counterstained and simultaneously mounted with Fluoromount G with DAPI (Invitrogen, Waltham, MA, USA). The slides were read under an epifluorescence microscope (Nikon Eclipse Ci, Amsterdam, Netherlands) at 40× magnification. Spermatozoa with intact acrosome were FITC+ and showed intense green fluorescence at the level of the acrosome, while spermatozoa with broken or lost acrosome were FITC– or with fluorescence at the level of the middle region of the head. 

### 2.9. Genomic Damage Analysis

Analysis of any DNA breaks was performed using a Halosperm Kit (Halotech, Madrid, Spain). Briefly, the sample was diluted to 20 mil/mL in PBS. Next, 50 µL of the sample was mixed with 100 µL of agar, previously dissolved for 5 min in a water bath at a temperature of 95–100 °C. An amount of 8 µL of the mixture was placed in the center of an agarized slide and covered with a coverslip of 22 × 22. The slide was transferred at 4 °C for 5 min to allow the agar to solidify. After removal of the coverslip, the sample was incubated with denaturing solution for 7 min and with lysis solution for 25 min. Next, the slide was incubated for 5 min in distilled water and then dehydrated by increasing alcohols (70% and 100% for 2 min each). Finally, staining was carried out by incubation with Eosin Y (Bio Optica, Milan, Italy) for 2 min, followed by Methylene blue (Bio Optica) for 2 min. The presence of a halo around the head was a hallmark of intact DNA. In fact, in the spermatozoa with fragmented DNA, the halo was absent.

### 2.10. MiOXYS (Male Infertility Oxidative System)

The MiOXYS is a system to assay the so-called static oxidation-reduction potential (sORP: static Oxidation-Reduction Potential) depending on the quantity of oxidants and antioxidants present in the sample tested. Following the supplier’s instructions, 30 µL of sample was placed on a sensor, previously inserted into the instrument, which, after approximately 2 min, provided a sORP value expressed in millivolt (mV). This value was then normalized by dividing it with the sperm concentration (mV/10^6^ sperm/mL). The cutoff considered in this experiment was proposed by Agarwal et al. (2017) [24]. It is called Youden’s Index and corresponds to 1.41 mV/10^6^ sperm/mL. Values above the threshold denote the presence of a general state of stress.

### 2.11. DCFH2-DA

ROS generation was monitored using of 2′,7′-dichlorofluorescein-diacetate (DCFH2-DA) probe. This compound cross membranes and, inside the cell, is deacetylated by intracellular esterases that reduce it to dichlorofluorescein (DCFH), a more hydrophilic and nonfluorescent molecule. In the presence of ROS, DCFH is rapidly oxidized into the fluorescent compound DCF. The stock solution of DCFH2-DA (130 µM) (Sigma-Aldrich) in DMSO was stored in aliquots at −20 °C until tested. If necessary, aliquots were thawed and diluted to a concentration of 13 µM in 150 µL of the sample. The samples were incubated for 30 min at 37 °C in the dark. Next, the spermatozoa were washed by adding PBS in a 1:1 ratio and centrifuged at 2000 rpm for 10 min. The supernatant was discarded, and the pellet was resuspended in 150 µL PBS. Cells were counterstained with 1 mM Hoechst 33342 (specific for nuclei) for 1 min, washed quickly in PBS, and observed under an epifluorescence microscope (Nikon Eclipse Ci). The acquisition of images was realized through “acquire multichannel images” tool, selecting DAPI and FITC filters. Spermatozoa with oxidative stress were FITC+/Hoechst+ (fluorescent in green and blue), while negative spermatozoa were FITC−/Hoechst+ (fluorescent in blue). The acquired images were analyzed using the software, which enabled the production of graphs of the fluorescence intensity emitted by each spermatozoon.

### 2.12. Cytoprotective Marker: HSP70 Expression

To investigate the stressogenic state of the cells and the stimulation of HSP70s protein expression following the exposure of human spermatozoa to PS-NPs, an immunohistochemistry protocol (qualitative analysis) was performed: 0.5 × 10^6^ spz/mL were fixed in 4% paraformaldehyde in PBS (1:1 ratio, *v*/*v*) for 15 min at room temperature. Next, the samples were centrifuged at 2000 rpm for 5 min, and after discarding the supernatant, the pellet was resuspended in PBS, twice. At this point, a drop of the sample was placed on a slide, smeared, and allowed to air dry. The cells were first permeabilized with PBS-0.3% Triton X-100 for 20 min and then covered with blocking solution consisting of PBS-3% BSA for 30 min. The slides were incubated with polyclonal anti-HSP70 primary antibody (rabbit, GeneTex, Irvine, CA, USA) diluted to 1:100 in PBS-3% BSA at 4 °C overnight. Following repeated washes in PBS, the slides were covered with Goat Anti-rabbit IgG pre-adsorbed Rhodamine (GeneTex) secondary antibody diluted to 1:100 in PBS-3% BSA for 1 h at room temperature. Finally, after subsequent washes in PBS, the slides were counterstained and simultaneously mounted with Fluoromount G with DAPI. To exclude possible autofluorescence of the samples, a negative control, in which the primary antibody was not added, was performed. The slides were analyzed by an epifluorescence microscope with 40× magnification, and the multichannel (DAPI and TRICT filters) images obtained were analyzed by software to evaluate the fluorescence intensity emanated for each individual spermatozoon.

### 2.13. Assessment of Mitochondrial Activity

3′-3′-Diaminobenzidine (DAB, Sigma-Aldrich) was applied to assess mitochondrial activity and function based on the deposition of DAB after the oxidation by cytochrome C oxidase. For this purpose, 100 µL of semen was incubated with 300 µL of 1 mg/mL solution of DAB in PBS for 1 h at 37° in the dark. Next, 10 µL of the suspension was smeared onto a slide, which was first allowed to air dry, and then fixed in 10% formaldehyde in distilled water for 10 min. Finally, the slides were washed in distilled water, allowed to air dry, and analyzed using a light microscope with 100× magnification under oil immersion. Observations permitted the classification of spermatozoa into four classes based on the staining of the intermediate segment:Class I: 100% of the intermediate segment stained.Class II: >50% of the intermediate segment stained.Class III: <50% of the intermediate segment colored.Class IV: uncolored intermediate segment.

### 2.14. Statistical Analysis

Past 4.0 Software was used to analyze the results and highlight potential statistically significant differences between the exposed groups and the control. Specifically, a one-way ANOVA test was performed, followed by a Tukey’s test. In addition, Pearson’s method was adopted to define the correlation coefficient between different parameters and protective response (expression of HSP70). Graphs were constructed using GraphPrism software. The level of significance was set as <0.05, and the data were indicated with the symbol * if significant (*p* < 0.05) and with the symbol ** if highly significant (*p* < 0.01). All data are presented as mean percentage ± standard deviation, excluding the normalized sORP values and the different characteristics of movement.

## 3. Results

### 3.1. Motility

As shown in Appendix A, PS-NPs with a larger diameter have not induced changes in the parameters describing the movement of spermatozoa. In contrast, in the case of the 50 nm PS-NPs, adverse effects are evident. Firstly, exposure has caused a reduction in the percentage of mobile spermatozoa in samples subjected to 0.5 µg/mL and 1 µg/mL. For the latter concentration, lower values in LIN and VCL and an upper value of WOB have been noted. A decrease in the LIN value (linearity of the curvilinear path) and the VCL value (distance travelled per second) indicates a reduction in the space travelled by the exposed spermatozoa. In addition, an increase in WOB (spermatozoa head movement) describes an intensification in head oscillation, a sign of difficulty of progression in space.

### 3.2. Membrane Integrity

The Eosin test differentiated spermatozoa with an intact membrane (viable and transparent) from spermatozoa with an injured membrane (non-viable and pink), as observed in Figure 1a. Figure 1b shows the trend of sperm with intact plasma membranes, following exposure to both PS-NPs at increasing concentrations. The rates of the spermatozoa exposed to the 100 nm PS-NPs were very similar to those of the control, while those of the cells exposed to the 50 nm PS-NPs showed lower values, especially for the highest concentration 1 µg/mL (16.52 ± 0.03, * *p* = 0.015) (Appendix A).

### 3.3. Acrosome Damage

The PNA-FITC test (Figure 2 and Figure 3) revealed no significant variation in the samples exposed at the lowest concentrations of both PS-NPs, while statistically significant values have been obtained for the highest concentration tested (1 µg/mL) for 50 nm (64.54 ± 0.01; *p* = 0.016 *) and 100 nm (65.9 ± 0.02; *p* = 0.02 *) PS-NPs compared to the control group (48.14 ± 0.03). All data are summarized in Appendix A.

### 3.4. DNA Fragmentation

DNA fragmentation was assessed using a Halosperm Kit, which differentiated spermatozoa with fragmented DNA (absence of halo) from gametes with intact DNA (presence of halo) (Figure 4a,b). Statistical analysis showed no significant changes between the different exposed groups and the CTRL, as can be seen in Figure 4c, regarding the PS-NPs with higher diameter. On the contrary, an elevated rate of DNA fragmentation has been found on the samples exposed to all concentrations of 50 nm PS-NPs −51.71 ± 0.02 (*p* < 0.001 **) for 0.1 µg/mL; 52.69 ± 0.01 (*p* < 0.001 **) for 0.5 µg/mL; 52.5 ± 0.02 (*p* < 0.001 **) for 1 µg/mL compared to the control group (36.64 ± 0.03) (Appendix A).

### 3.5. MiOXYS Analisys

MiOXYS was used to analyze the total balance of oxidant and antioxidant systems present in the sperm. As shown in Figure 5, an increment in oxidative stress has been observed for both PS-NPs, even though the worst result was obtained in the samples exposed to the 50 nm PS-NPs at the highest concentration (1 µg/mL) (Appendix A). 

### 3.6. ROS Production

The general higher stress has been correlated with an overproduction of ROS in the exposed samples. As shown in Figure 6a–c, through DCFH2-DA probe, spermatozoa have been categorized into two groups: with overexpression of ROS (DCF+/Hoechst+) or with absence of ROS (DCF−/Hoechst+). It can be seen from Figure 7a,b that the increase in ROS has occurred in the samples exposed to both PS-NPs at all concentrations. Again, the most significant alteration has been in the higher concentration of the smaller PS-NPs (Appendix A).

### 3.7. HSP70s Expression

Qualitative immunohistochemistry analysis detected the synthesis of HSP70s proteins only in the samples exposed to the smaller 50 nm PS-NPs, in which a highly significant (*p* < 0.01 **) (Appendix A) increase in their expression has been found, mainly at the level of the neck, and a smaller portion of spermatozoa, throughout flagellum, as observed from Figure 8a–d and Figure 9. No positivity has been shown in the control samples and in all samples exposed to the increasing concentrations of the 100 nm PS-NPs. From the Pearson coefficients obtained, a negative correlation has emerged between the marked expression of HSP70s and the reduction in DNA fragmentation (r = −0.968, r^2^ = 0.937, *p* =< 0.001 **), oxidative stress (r = −0.965; r^2^ = 0.932, *p* =< 0.001 **), and mitochondrial dysfunction (r = −0.982, r^2^ = 0.965, *p* =< 0.001 **), as shown in Figure 10.

### 3.8. Mitochondrial Activity

DAB was used to analyze the status and function of mitochondria (Figure 11). From Figure 12a,b, following exposure to the 50 nm PS-NPs compared with the control, it can be seen that the portion of sperm belonging to class I (with excellent mitochondrial functionality) has diminished in all exposed samples, especially for the higher concentration, and the percentage of sperm belonging to class IV (lack of functionality) has increased. On the contrary, in the case of samples exposed to the PS-NPs with a larger diameter (100 nm), no significant differences are evident (Appendix A). 

## 4. Discussion

Nanoplastics pollution has generated growing apprehension in the academic community, especially because the most recent and alarming findings have corroborated the ability of nanoplastics to penetrate the human body and be absorbed and biodistributed through the bloodstream to bioaccumulate in different organ. Because several studies on mouse models have confirmed the sensibility of the testis to the influence of nanoplastics, proposing and translating this result also for other mammals, including humans, in the present study, solutions containing amino-modified PS-NPs at increasing concentrations have been tested on human spermatozoa to assess, after 30 min of exposure, the fundamental parameters of the semen. Given the absence of quantification studies of the exact concentrations accumulated in the human testis as a result of distribution by the bloodstream, the concentrations tested could be higher than realistic. However, the goal of the present experiment was to detect and quantify the potential impact of NPs on the metabolism and structure of human sperm cells, identifying potential toxic concentrations. 

### 4.1. Motility

In general, 50 nm PS-NPs exerted more negative effects than 100 nm PS-NPs, demonstrating how in vitro toxicity is associated with the dimension of the particles. First, the 50 nm PS-NPs, in contrast to 100 nm PS-NPs, not only resulted in a reduction in the percentage of mobile spermatozoa but have also caused a decreased in the speed and in the linearity of the pathway (LIN). Additionally, an increase in the lateral oscillations of the spermatozoa head (WOB), as a consequence of the decline in VCL, has been observed. Impaired motility has always been correlated with dysfunction, as also evidenced by our results, at the mitochondrial level, at the site of ATP production. The picture emerging from the literature, however, appears to be more complex because ATP derived from mitochondria appears to be flanked by ATP produced by glycolysis or by oxidation of endogenous substrates. Thus, in the resulting reduction in motility, parallel to the worsening in mitochondria functionality, other mechanisms, not currently investigated, could be involved [25]. 

### 4.2. Alteration of Plasmatic and Acrosomial Membranes

Our study has also shown an alteration of the acrosome morphology and a significant compromission of the membrane. The worst results were recorded for 50 nm PS-NPs, which have also caused plasmatic membrane insults, compared to the larger PS-NPs and to the control. Given the scant data in the literature concerning the possible interaction of nanoparticles with human spermatozoa, the results reported are certainly preliminary, but they provide additional information to evidence other studies carried out on murine models. Jin and et al. (2021), in fact, emphasized the susceptibility of acrosome to the action of PS-NPs, which caused the disaggregation and the loss of this vesicle [14]. Furthermore, the different responses observed seem to be attributable exclusively to the size of the nanoparticles, the only difference between the two groups of samples tested. The limited data in the literature focus on the spermatozoa of aquatic species (*Crassostrea gigas*), specifically focusing on the different response of spermatozoa to amino- and carboxy-modified NPs and not in relation to size. However, considering the different papers, a high spermiotoxicity (characterized by a decrease in the percentage of motile spermatozoa (−79%) and in the velocity (−62%) compared to control spermatozoa) was found in samples exposed to 50 nm NPs from concentrations at 10 µg mL^−1^ up to 25 mL^−1^ [26]. González-Fernández et al. (2018), on the other hand, obtained no increase in ROS production in samples exposed to different increasing concentrations (from 0.1 to 100 mg L^−1^) of 100 nm NPs [27]. Although these data concern species very distant from humans, it is important to consider that the spermatozoon is a highly conserved cell during evolution and that many of the mechanisms at the basis of its physiology are shared with all species. This highlights the importance of conducting further studies to expand our knowledge of the possible variation in the toxicological profile of a substance depending on its size.

### 4.3. DNA Fragmentation: Possible Correlation with ROS and Mitochondrial Disfunctions

In our study, negative outcomes have also been noted for DNA fragmentation, which is significantly increased in samples exposed to the smaller PS-NPs. This could be due to the direct action of particles on the DNA molecule or to a secondary effect related to the overproduction of ROS. The insults to DNA, and also to mitochondria, by 50 nm PS-NPs suggest their possible uptake by spermatozoa, unlike PS-NPs with larger diameters. Although the mechanism underlying the interaction between PS-NPs and membranes remains to be clarified, several mechanisms have been proposed to explain the internalization. The most plausible one concerns the possibility of PS-NPs with a not excessively large size (40–60 nm) to penetrate inside cells via endocytosis occurrence [28], and also to afflict various organelles, moving to the level of the nucleus (DNA fragmentation) and mitochondria. Larger PS-NPs, on the other hand, could affect sperm cells while not being internalized, for instance by interacting with receptors present at the level of the plastic and acrosomal membranes.

For instance, this might explain why both PS-NPs have altered the stress, as shown by sORP values, indices of the overall state of the cell (considering all oxidant and antioxidant species present on spermatozoa), although it seems clear that 50 nm PS-NPs can also interact with internal organelles. On sperm cells exposed to the smaller PS-NPs, oxidative stress has mainly been provoked by an overexpression of reactive oxygen species (ROS), apparently correlated with mitochondrial dysfunctions. It is well known that oxidative stress and mitochondrial anomalies are intimately related in a vicious circle, in that unstable species can attack and degrade mitochondria membranes, while the latter, because of the damage and subsequent malfunction, release additional reactive species [29]. The effects of oxidative stress, as confirmed in the study by Bisht et al. (2017), can have serious repercussions on sperm function. In fact, spermatozoa are highly vulnerable due to limited levels of antioxidant defense, and high levels of oxidative stress can damage sperm DNA, RNA transcripts, and telomeres and, therefore, may provide a common etiology underlying male infertility and recurrent pregnancy loss, as well as congenital malformations and complex neuropsychiatric disorders [30]. It remains to be clarified which unstable molecules are involved in the altered oxidative stress in samples exposed to 100 nm PS-NPs. The involvement of radicals such as NO in the proper physiological functions of the spermatozoon is well known. The balance of oxidant and antioxidant complexes, again, has been unbalanced, tending towards the predominance of unstable and harmful compounds other than ROS [31].

### 4.4. Cytoprotective Responses: Expression of HSP70s

The synthesis and localization of certain protective proteins, HSP70s, have been qualitatively analyzed. HSP70s molecules can be constitutive or inducible by various stimuli, including stress. They, in fact, act as chaperonins by protecting proteins from denaturation and preventing the formation of aggregates between misfolded molecules by eliminating them or assisting them to achieve natural folding [32]. Another cytoprotective mechanism concerns the ability of HSP70s to protect and repair DNA breaks following ROS insults, complementing the cell’s antioxidant systems [33]. The present study has revealed that increased HSP70s activity and their concentration at the level of the intermediate segment, in samples exposed to 50 nm PS-NPs, have been negatively correlated with DNA breaks, oxidative stress, and mitochondrial dysfunctions; thus, samples with less damage have exhibited a higher percentage of spermatozoa expressing HSP70s. Interestingly, spermatozoa exposed to both types of PS-NPs possess high sORP values compared to the control, but only those exposed to the smaller PS-NPs have shown HSP70s production at the level of the neck or in the entire flagellum. The inductive phenomenon of HSP70s synthesis is probably the overexpression of ROS (evidenced only in samples exposed to 50 nm PS-NPs) and not the eventual general state of stress of the cell. The defense mechanism could be associated with the protection of proteins and membranes of mitochondria and of DNA molecules through direct action on ROS by ensuring their proper activity.

Correct expression and proper functioning of HSP70s appear to be responsible for the ability of sperm cells to adapt, even to unfavorable microenvironmental conditions. Low levels of HSP70, in fact, have been observed in infertile patients [34]. Despite this, it remains to be clarified how and in what timeframe HSP70s act to remove the stressogenic chemical species from the gametes and whether this defense mechanism can bring stress levels back within ranges considered normal and physiological.

### 4.5. Environmental and Biomedical Implication

The chronic persistence of plastic waste in environmental matrices has required careful monitoring of the timing and different ways of its degradation, as well as its effects after penetration into organisms [35]. As the ubiquitous nature of NPs has been confirmed, all organisms are subject to contact with them by inhalation, ingestion, or absorption [36]. 

The reproductive sphere seems to be the most impacted by plastics, as gametes are highly specialized cells, sensitive to alterations in their microenvironment, whether released in water (internally fertilized species), or in the female genital tract (internally fertilized species).

Gametic quality is, in fact, correlated with reproductive success, so any perturbation that affects the optimal parameters for successful fertilization inevitably leads to a reduction in the organisms of a population and very often to an imbalance in entire ecosystems.

In mammals, the ability of NPs to cross the hemato–testicular barrier certainly confronts the scientific community with a problem that has perhaps not been adequately addressed until now [37]. NPs, in fact, could contribute to the deterioration of gametic quality. Moreover, because fertilization is a process that takes place in the female genital tract, the presence of plastic residues could partly explain idiopathic infertility. In this context, a man with normospermic parameters could release gametes into the female genital tract to achieve pregnancy, but if spermatozoa encounter pollutants in their trajectory, such as NPs, for example released from period products normally and widely used by women, their physiology could be adversely affected, leading to couple infertility [38]. 

Thus, it is evident that more knowledge is needed on the real impact of the use of plastic products on organisms and ecosystems. It should be considered important to study the concentrations capable of reaching mature spermatozoa, both in the testis and in the female genital tract, to determine the effects on egg cells as well and, finally, to assess the effects on fertilization. 

## 5. Conclusions

These findings contribute to the elucidation of responses of human spermatozoa to the presence of high concentrations of PS-NPs in their microenvironment, highlighting metabolic and structural anomalies of essential components and organelles. The toxic effect appears to be correlated with the dimension of the nanoparticles, because worst results have been obtained for PS-NPs with smaller diameters. Finally, after the impact of PS-NPs, spermatozoa respond to the increased production of ROS with the synthesis of HSP70, which actively participates in damage recovery.

## Figures and Tables

**Figure 1 biology-12-00624-f001:**
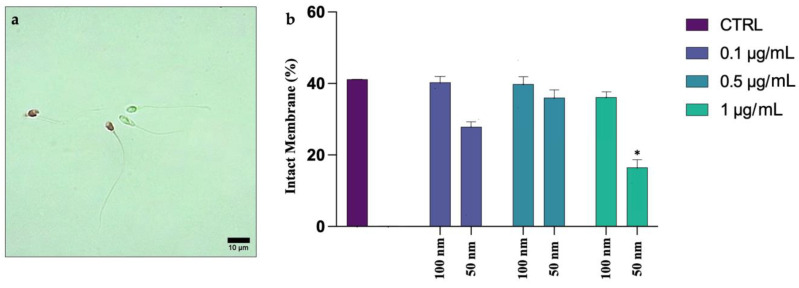
(**a**) *Eosin* test: spermatozoa with intact membrane (translucid) and spermatozoa degraded membrane (pink); (**b**) mean percentage of spermatozoa with intact membrane after exposure (30 min) to 50 and 100 nm PS-NPs at increasing concentrations. Approximately 400 spermatozoa were counted in five different fields for each sample. In addition, two replicates were performed, in a total of 8 samples and 3200 spermatozoa analyzed. Statistic differences (* *p* < 0.05) have been observed between group exposed to 1 µg/mL of 50 nm PS-NPs and control (CTRL).

**Figure 2 biology-12-00624-f002:**
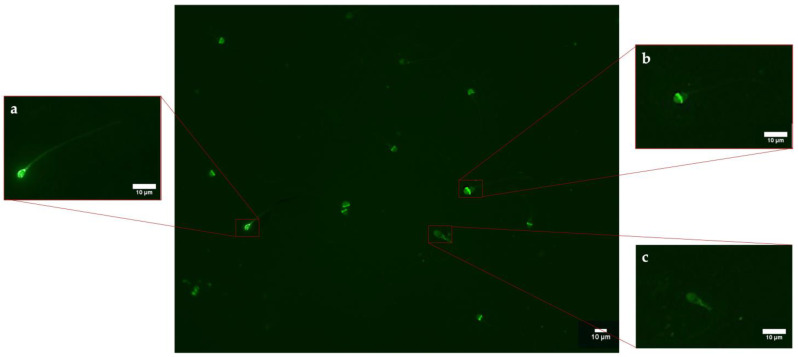
Observation of a casual field of spermatozoa subjected to PNA-FITC test to assess acrosome integrity: (**a**) spermatozoon FITC+ with intact acrosome; (**b**) FITC+ spermatozoon only at the level of the middle region of the head; (**c**) spermatozoa FITC−.

**Figure 3 biology-12-00624-f003:**
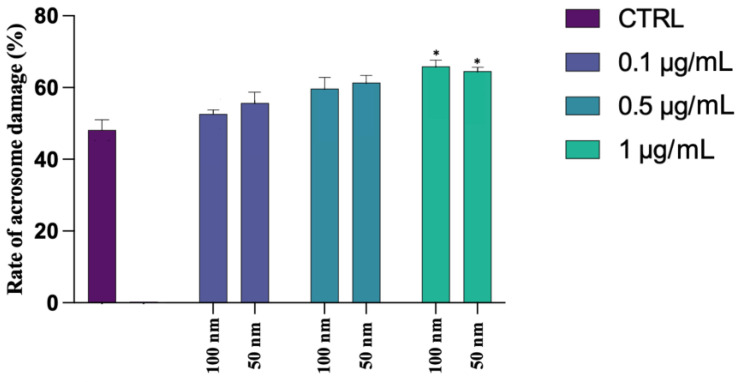
Mean percentage of spermatozoa with acrosome damage after exposure (30 min) to 50 and 100 nm PS-NPs with increasing concentrations. Approximately 400 spermatozoa were counted in five different fields for each sample. In addition, two replicates were performed, in a total of 8 samples and 3200 spermatozoa analyzed. Significant data have been found at the higher concentration of both PS-NPs (* *p* < 0.05).

**Figure 4 biology-12-00624-f004:**
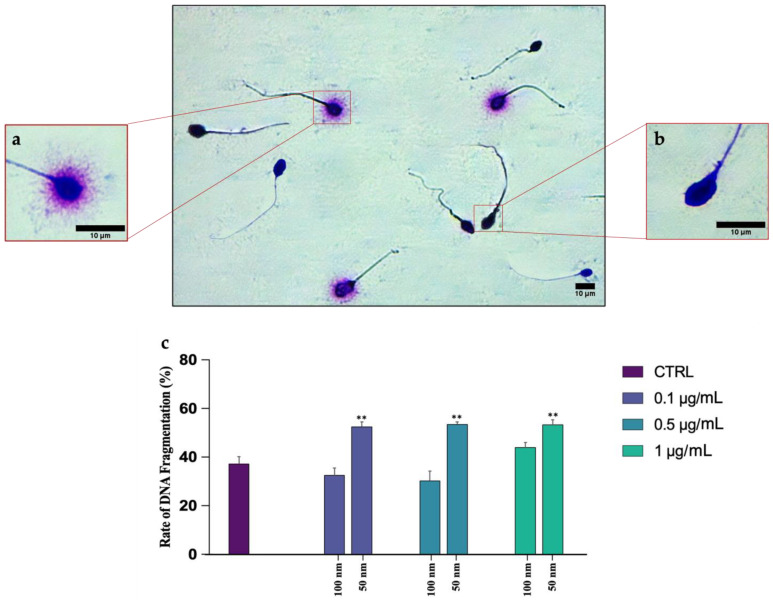
Observation of a casual field of spermatozoa subjected to halo test to assess DNA fragmentation: (**a**) spermatozoon with intact DNA surrounded by a halo; (**b**) spermatozoon with fragmented DNA without halo; (**c**) mean percentage of spermatozoa with fragmented DNA after exposure (30 min) to 50 and 100 nm PS-NPs at increasing concentrations. Approximately 400 spermatozoa were counted in five different fields for each sample. In addition, two replicates were performed, in a total of 8 samples and 3200 spermatozoa analyzed. Strong statistically differences (** *p* < 0.01) have been found between samples exposed to all concentration of 50 nm PS-NPs compared to the control group.

**Figure 5 biology-12-00624-f005:**
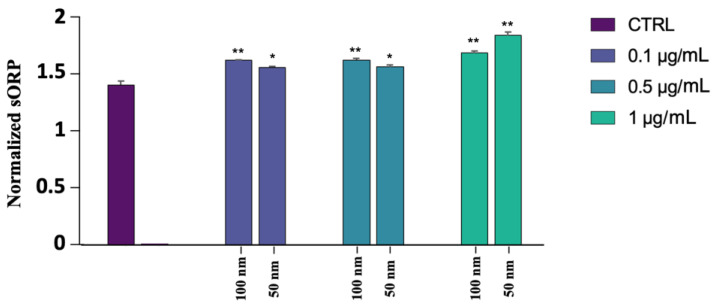
Normalized sORP values of spermatozoa exposed to 50 and 100 nm PS-NPs at increasing concentrations. Strong statistically differences (** *p* < 0.01) have been found between samples exposed to all concentration of 100 nm PS-NPs and 1 µg/mL 50 nm PS-NPs compared to the control group. Significant data (* *p* < 0.05) have been found at 0.1 µg/mL and o.5 µg/mL of 50 nm PS-NPs.

**Figure 6 biology-12-00624-f006:**
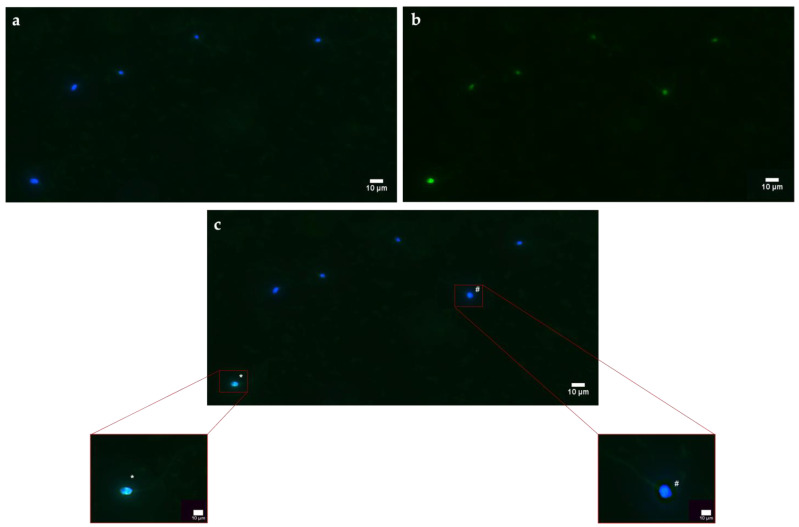
Observation of a casual field of spermatozoa subjected to DCFH2-DA test to assess ROS production: (**a**) observation of spermatozoa under epifluorescence microscope with 40× magnification, using DAPI channel acquisition; (**b**) observation of spermatozoa under epifluorescence microscope with 40× magnification, using FITC channel acquisition; (**c**) observation of spermatozoa under epifluorescence microscope with 40× magnification, using multichannel acquisition. The symbol * indicates a spermatozoon DCF+/Hoechst+, while # indicate a spermatozoon DCF−/Hoechst+.

**Figure 7 biology-12-00624-f007:**
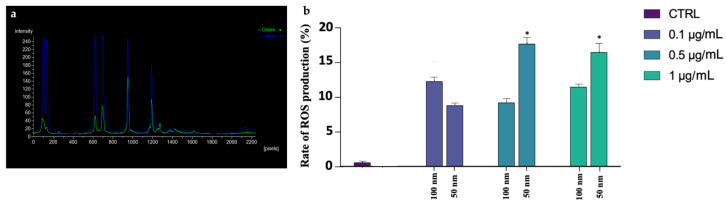
(**a**) Analysis of fluorescence intensity of each spermatozoon. Blue peaks indicate the intensity of Hoechst 33342 that is specific for the nucleus, while green peaks show spermatozoa positive to the DCFH2-DA probe; (**b**) mean percentage of spermatozoa positive of oxidative stress after the exposure (30 min) to 50 and 100 nm PS-NPs increasing concentrations. Approximately 400 spermatozoa were counted in five different fields for each sample. In addition, two replicates were performed, in a total of 8 samples and 3200 spermatozoa analyzed. Significant data were obtained at the two higher concentration (0.5 µg/mL and 1 µg/mL) for 50 nm PS-NPs (* *p* < 0.05).

**Figure 8 biology-12-00624-f008:**
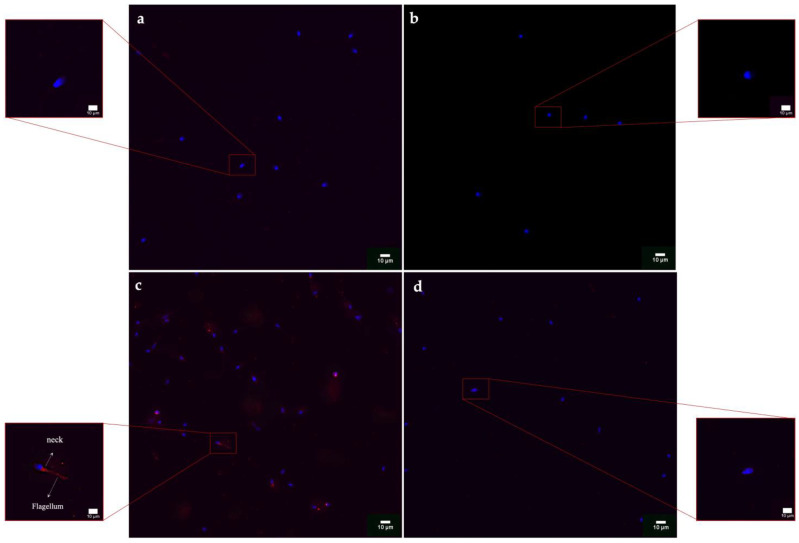
Results of immunohistochemistry protocol for the identification of HSP70 after the exposure (30 min) to 50 and 100 nm PS-NPs at increasing concentrations: (**a**) control (CTRL) with a magnification of a spermatozoon without red spots; (**b**) sample exposed to 100 nm PS-NPs with a magnification of a spermatozoon without red spots; (**c**) sample exposed to 50 nm PS-NPs with a magnification of a spermatozoon with red fluorescence diffused on flagellum; (**d**) negative control with a magnification of a spermatozoon without red spots.

**Figure 9 biology-12-00624-f009:**
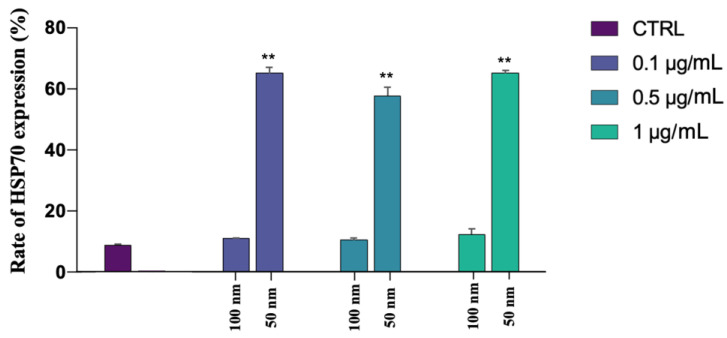
Mean percentage of spermatozoa that expressed HSP70 after the exposure (30 min) to 50 and 100 nm PS-NPs at increasing concentrations. Approximately 400 spermatozoa were counted in five different fields for each sample. In addition, two replicates were performed, in a total of 8 samples and 3200 spermatozoa analyzed. Strong statistically differences (** *p* < 0.01) have been found between samples exposed to all concentration of 50 nm PS-NPs compared to the control group.

**Figure 10 biology-12-00624-f010:**
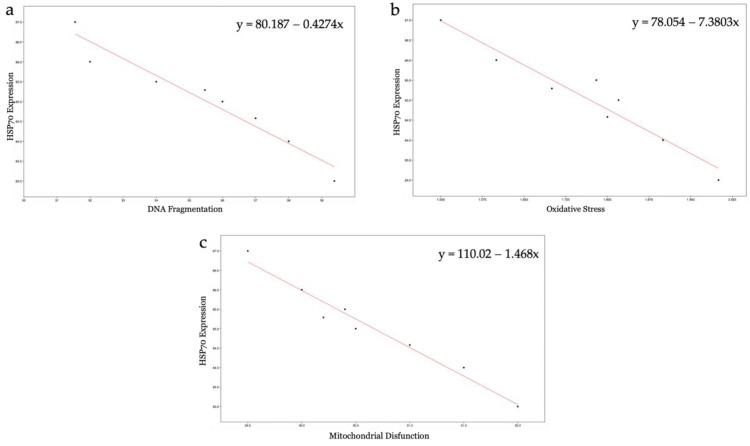
Correlation analysis: (**a**) relation between HSP70 expression and DNA fragmentation; (**b**) relation between HSP70 expression and oxidative stress; (**c**) relation between HSP70 expression and mitochondrial disfunction.

**Figure 11 biology-12-00624-f011:**
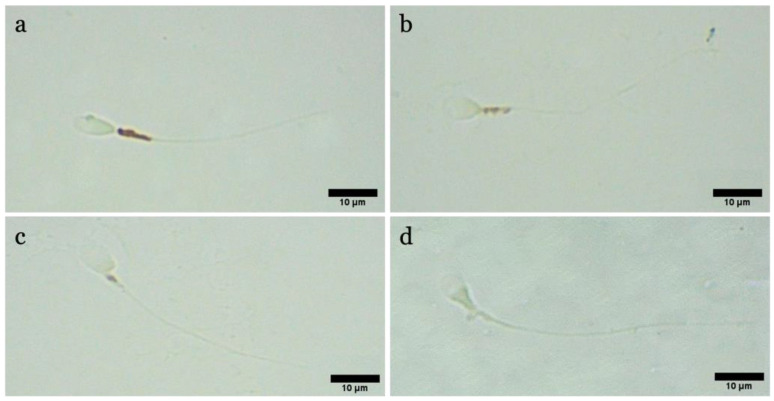
Observation of a casual field of spermatozoa subjected to DAB test to assess mitochondrial activity (DAB): (**a**) spermatozoon of class I with 100% of intermediate segment colored; (**b**) spermatozoon of class II with >50% of intermediate segment colored; (**c**) spermatozoon of class III with <50% of intermediate segment colored; (**d**) spermatozoon of class IV with no colored intermediate segment.

**Figure 12 biology-12-00624-f012:**
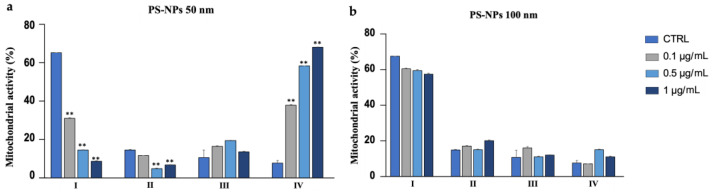
(**a**) Mean percentage of mitochondrial activity of spermatozoa after exposure (30 min) to 50 nm PS-NPs at increasing concentrations. Strong statistically differences (** *p* < 0.01) have been found between samples and control groups; (**b**) Mean percentage of mitochondrial activity of spermatozoa after exposure (30 min) to 100 nm PS-NPs at increasing concentrations. Approximately 400 spermatozoa were counted in five different fields for each sample. In addition, two replicates were performed, in a total of 8 samples and 3200 spermatozoa analyzed. No statistical differences have been found between samples and control groups.

**Table 1 biology-12-00624-t001:** Parameters fresh semen.

Parameters	Mean ± SD
Age (years)	31 ± 8.39
Volume (mL)	3.1 ± 0.7
Concentration (10^6^ sperm/mL)	121.83 ± 48.13
Progressive motility (%)	60 ± 9.63
Non progressive motility (%)	7.66 ± 3.26
Immobility (%)	32.83 ± 9.41
Normal Morphology (%)	8.16 ± 3.06

Data are showed as mean ± standard deviation (SD).

**Table 2 biology-12-00624-t002:** Parameters measured by CASA plugin.

CASA Parameters	
Percent motility	Percent of motile spermatozoa.
Velocity Curvilinear (VCL)	Total distance traveled per second.
Velocity average path (VAP)	Point to point velocity on a path constructed using a roaming average. The number of points in the roaming average is 1/6th of the frame rate of video used.
Velocity straight line (VSL)	Velocity measured using the first point and the average path and the point reached that is furthest from this origin during the measured time period.
Linearity (LIN)	VSL/VAP, describes path curvature.
Wobble (WOB)	VAP/VCL, describes side-to-side movement of the sperm head.
Beat cross frequency (BCF)	This value is determined in the plugin by detecting the frequency at which VCL crosses VAP.

**Table 3 biology-12-00624-t003:** Optimal values entered in the CASA plugin to correctly analyze the different movement parameters of human spermatozoa.

Parameters	
a. Minimum sperm size (pixel)	1
b. Maximum sperm size (pixel)	250
c. Minimum track length (frames)	20
d. Minimum sperm velocity between frames (pixel)	20
e. Minimum VSL for motile (µm/s)	3000
f. Minimum VAP for motile (µm/s)	20
g. Minimum VCL for motile (µm/s)	25
h. Low VAP speed (µm/s)	5
i. Maximum percentage of path with zero VAP	1
j. Maximum percentage of path with low VAP	25
k. Low VAP speed 2 (µm/s)	25
l. Low VCL speed (µm/s)	35
m. High WOB (percent VAP/VCL)	80
n. High LIN (percent VSL/VAP)	80
o. High WOB two (percent VAP/VCL)	50
p. High LIN two (percent VSL/VAP)	60
q. Frame Rate (frames per second)	59
r. Microns for 1000 pixel	1075

## Data Availability

Original data are available on request.

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
