# Peer review of "In Vitro Nano-Polystyrene Toxicity: Metabolic Dysfunctions and Cytoprotective Responses of Human Spermatozoa"

_biology, 2023, doi:10.3390/biology12040624_

Round 1
Reviewer 1 Report
Martina Contino reported an interesting study on the toxicity of PS NPs on human spermatozoa. The topic was of significance, and the manuscript fell within the scope of Biology. However, before a second consideration, a Major Revision was suggested by the reviewer. Following are the detailed comments.
(1) A Materials Section should be supplemented under Section 2.1 to include the main materials.
(2) The significant digits in Table 1 should be unified.
(3) Scale bars must be added in micrographs like Figure 1a, 2a, 2b, 3a, etc.
(4) For intact membrane, acrosome damage, DNA fragmentation, etc., the Control group did not “behave normally”. For instance, the Control group had only 40% intact membrane. So was it a positive control? The reader might tacitly approve that the Control group was a negative control.
(5) The resolution of Figure 6 was low. Please consider to replace it with a high-quality one.
(6) Only two sizes (50 and 100 nm) were included in the tests. It was actually a bit premature to conclude the correlation between size and toxicity. Please make careful discussion upon this issue.
(7) In the Discussion Section, it was advisable to divide it into several subtitles. In addition, the environmental or biomedical implications of these findings should be envisioned.
(8) It should be noticed that ‘Funding Section’ and ‘Acknowledgement Section’ conveyed contradictory information.
(9) Only 5 out of 27 papers cited were published before 2020. Please notice that Biology was a flagship journal in its field, and more recent papers deserved referenced.
Author Response
Dear Revisor,
I am sending you the revised manuscript based on your suggestions.
Sincerely,
Martina

Reviewer 2 Report
Dear authors,
I enjoyed reading this MS, in which sperm cells were exposed to nanoplastics and a set of responses were then evaluated. I suggest some minor revisions, that I think can improve you manuscript before it is published. These specific comments are listed below.
Lines 75-77: Please check for grammar and clarity.
Line 79: remove comma after Jin et al.
Line 93: Please provide more details on how the stock solution was prepared. Which liquid was used to disperse the plastics? As nanoplastics may agreggate, have you used sonication? if so, please add any description.
Line 95: Do the authors have any approximated estimation of how many particles are present in 1, 0.5 and 0.1 micrograms of NPs? This is relevant to evidence if the test-concentrations are realistic (I am guessing that the concentration used is much higher than those reported, thus it must be clearly explicited in the text if was the case). I highlight that using high concentrations can be justified by the search for effective concentrations.
Lines 111-112: Please explain (here or in the section 2.4) if freezing and thawing the sperm cells could affect their healthy and viability.
Table 1: Include the unity for Age
Section 2.3: Please include the due authorizations as supplementary material
Lines 139-142: It is not clear to me how the sperm cells were exposed to the NPs dispersions. Was the exposure made in glass tubes, or microplates, or in the microtubes, or another type of chamber? Please provide some more details.
Lines 151-152: This table has no legend. Moreover, it is a bit confuse if all the parameters indicated in this table and in table 2 were taken and analyzed.
Fig. 10. The text regarding axis x is too small and cannot be read.
Line 383 and on (Discussion section): if the tested NP concentrations are higher than realistic ones, this must be addressed in some place of the discussion, and the readers need to be informed that this MS's goal is to detect and quantify the potential impacts NPs may cause on human sperm cells.
Lines 473-474: Considering my previous comment, maybe the authors need to change this statement into "These findings contribute to the elucidation of responses of human spermatozoa to the presence of high concentrations of PS-NPs in their microenvironment,"
Author Response

(The authors gave the same response as above.)

Reviewer 3 Report
Several environmental pollutants have an adverse effect on human spermatozoa and overall human reproductive health. Here, Contino et al. address an important question of how polystyrene nanoparticles (PS-NPs), one of the major environmental pollutants, impact human spermatozoa in vitro. Although more vigorous analysis is required to assess the impact on overall reproductive biology, this manuscript (MS) is an important step forward in understanding the impact of PS-NPs on human reproduction. Contino et al. investigated the toxicity of 50 and 100 nm PS-NPs on human sperm by monitoring different parameters like motility, membrane integrity, mitochondrial activity, DNA fragmentation, Acrosome integrity, and ROS production. The authors also correlate the higher expression of HSP70 protein as a possible mechanism to counteract the adverse effect imparted by PS-NPs. While this is a good study, I recommend significant improvements in data quality, presentation, and a better explanation of all the results. This will make MS look much better, and it would be much easier for readers to follow the significant findings in this MS.
Major Comments/Suggestions
1) None of the images are annotated, colors are not defined, and most are of poor quality.
2) The microscopy images are too small in Fig 2, 5, and 6. The images have a large field of view with only a few numbers of cells. Please provide low-magnification images to cover multiple cells and crop individual cells, and enlarge them to show different results.
3) Indicate the number of cells counted and the number of replicates for each test in the legend of all the figures. If possible, please include the exact number of cells out of the total cells for each result. This would give a clear picture of the sample size and reproducibility of the results.
4) For motility analysis, can authors provide the videos of control and treated sperm samples as supplementary files to compare the motility?
5) In Fig 2a and 2b, mark the acrosome and equatorial region with an arrow and enlarge the cell size to visualize the sperm head clearly.
6) In Fig 3, can authors include an image with multiple cells on the same field of view? Annotate the image and mark the cell with a halo and no halo.
7) In Fig 5a, it is very hard to see FITC+ cells. It looks like there is no FITC+ cell in this image. Please include representative images with FITC+ and FITC- cells. Also, enlarge to images. If necessary, include two panels to show control cells treated cells.
8) In the HSP70 expression analysis, the authors have claimed that the HSP70 expression is localized to the intermediate segment and flagellum. From the images provided, it is very hard to see the HSP70 signal in the intermediate segment or even in the flagellum. On the contrary, the HSP70 signal might be localized only at the neck region. Please include an enlarged image, clearly annotate, and mark the location to show HSP70 localization.
Minor Comments /Suggestions
1) Define various motility parameters used in CASA in the method section and line 274.
2) In lines 453-454, the authors stated that the HSP70 localized to the intermediate region of sperm. Please reevaluate the images to figure out the exact location.
Author Response

(The authors gave the same response as above.)

Round 2
Reviewer 1 Report
I have no further concerns.
Reviewer 3 Report
Thank you for addressing all my concerns.